# END-TO-END QA CONSTRUCTION PIPELINE FOR CONTINUAL PRE-TRAINING OF LARGE LANGUAGE MODELS

## ABSTRACT

As Large Language Models (LLMs) evolve into proficient AI assistants, the demand for high-quality data becomes increasingly critical. Existing methods to create question-answer (QA) datasets often depend on limited self-generated data from LLMs or labor-intensive manual annotations, which restrict both the scope and size of the resulting datasets. To overcome these challenges, we propose a comprehensive pipeline for acquiring and filtering high-quality QA data from web searches, utilizing the vast and diverse content available online. Our approach includes training the High-Quality Knowledge Model, which ensures dataset robustness by filtering queries based on clarity and static knowledge criteria. Additionally, we introduce the Knowledge Boundary Model to pinpoint and address knowledge boundaries within LLMs, enhancing their ability to manage novel scenarios effectively. Our approach not only results in the generation of an extensive QA dataset but also implements training strategies that boost LLM capabilities. Our method improves the baseline by 22.96% on Chinese SimpleQA, 4.66% on SimpleQA, 4.78% on seven single-hop datasets, and 17.47% on eight multi-hop datasets. Our code and data will be released.

## 1 INTRODUCTION

As Large Language Models (LLMs) gradually demonstrate their potential as advanced AI assistants, the need for vast amounts of high-quality data becomes paramount (Radford et al., 2019; Brown et al., 2020; Touvron et al., 2023; Dubey et al., 2024). This necessity is driven by effective training, where the quantity and quality of the training dataset play a pivotal role (Li et al., 2023a; Brown et al., 2020; Chowdhery et al., 2023; Touvron et al., 2023; Dubey et al., 2024). Compared to general text data, question-answer (QA) data can better enhance a model's ability to identify and address knowledge gaps, similar to how humans consolidate learning through practice problems. Existing methods for creating QA datasets frequently use LLMs for self-improvement on limited datasets (Xu et al., 2023; Lewkowycz et al., 2022), or rely on manually annotated data (Brown et al., 2020; AI@Meta, 2024; Li et al., 2023b), which is time-intensive and requires substantial effort. Additionally, these sources tend to be restricted in both scope and size.

To address this challenge, we propose a pipeline for acquiring and filtering high-quality QA data from web searches. Web contents offer a wealth of diverse, high-quality data across various domains. Therefore, our pipeline initially crawls web content from sites like Wikipedia and Baidu. It then uses LLMs to construct queries from the web content. To obtain high-quality queries, we trained an High-Quality Knowledge Model specifically used to filter queries based on their clarity, static nature, and knowledge basis. This model effectively weeds out ambiguous, ephemeral, or opinion-based questions, ensuring the dataset's longevity and reliability. Afterward, we use Google search to acquire knowledge relevant to the questions and answers. We then input the search content and questions to the LLM to generate responses using a retrieval-augmented generation (RAG) method. Finally, we implement a refusal filtering process to ensure quality. During this

phase, generated answers are validated to confirm that they provide comprehensive and pertinent responses to the original queries. As a result, from the initially generated QA dataset of 26B tokens, we filtered out a high-quality QA dataset consisting of 18B tokens.

Recognizing that simply reinforcing known knowledge offers limited developmental gains, our framework incorporates a Knowledge Boundary Model to identify knowledge boundaries within language models. By systematically evaluating the consistency of model-generated answers to known queries, we categorize them into *known* and *unknown*. This distinction enables us to channel training efforts towards resolving knowledge gaps, thereby advancing model performance in handling novel or complex scenarios. Our approach achieves significant improvements over the baseline: 22.96% on Chinese SimpleQA and 4.66% on SimpleQA. It also demonstrates robust gains across multi-hop and single-hop QA datasets, with improvements of 17.47% (eight multi-hop datasets) and 4.78% (seven single-hop datasets), respectively.

Our contributions are as follows:

1. We introduce a comprehensive pipeline for acquiring and filtering high-quality QA data from web searches. This pipeline employs the Knowledge Boundary Model to enhance the model's knowledge of unknown areas, and we thoroughly validate its effectiveness through extensive experimentation.

2. Through our pipeline, we generated an extensive high-quality QA dataset consisting of 18B tokens. We will release this corpus to the research community to facilitate and advance further studies.

3. Our method achieves consistent improvements over the baseline across Chinese SimpleQA, SimpleQA, and multiple single-hop and multi-hop QA datasets.

## 2 RELATED WORKS

**Data Pipelines for LLMs.** The rise of LLMs has led to efforts focusing on building larger-scale and higher-quality datasets from web content to aid training. For instance, The Pile (Gao et al., 2020) used jusText (Endrédy & Novák, 2013) to extrapolate text from web content, creating Pile-CC. LLaMA (Touvron et al., 2023) adapted the CCNet pipeline to produce a vast close-sorced pre-training dataset. RedPajama (Computer, 2023) subsequently replicated LLaMA's dataset and made it publicly accessible. Advancing data quality further, RedPajama v2 (Computer, 2023) introduced 46 distinct quality metrics for multi-dimensional data characterization. RefinedWeb (Penedo et al., 2023) applied content extraction techniques on HTML documents from Common Crawl, obtaining cleaner, higher-quality text with a limited amount was shared publicly. In response, FineWeb (Penedo et al., 2024) replicated RefinedWeb, released the data publicly, and developed a filtering strategy to omit educational content, creating the FineWeb-edu pre-training dataset. DCLM (Li et al., 2024) extracted extensive textual data from web content and crafted a tailored filter to gather a substantial body of instruction-style data, enhancing its quality significantly. Lastly, Redstone (Chang et al., 2024) introduced an efficient data pipeline focusing on general, code, math, and QA data by simplifying processing and expanding dataset size. However, past approaches used a unified model to handle all data, even though the data on the internet is vast and varied, and the patterns required for data from different sources are obviously different. Unlike past methods such as Redstone (Chang et al., 2024), which manually designed different filtering matches for general, code, math, and QA data, we train two filtering models and leverage the rich priors of LLMs to learn how to extract valuable knowledge from the vast content on the internet.

**QA data pipeline for LLMs.** Interactive QA capacities are fundamental to the applications of LLM. Yet, current approaches to developing QA datasets commonly depend either on LLMs for limited dataset self-improvement (Xu et al., 2023; Lewkowycz et al., 2022), or on manually annotated data (Brown et al., 2020; AI@Meta, 2024; Li et al., 2023b), which is both time-consuming and labor-intensive. Moreover, these methods are often constrained in terms of the scope and size of the generated data. Consequently, there is an

Figure 1: The pipeline for acquiring and filtering high-quality QA data involves multiple steps: web scraping to gather content, query extraction to generate questions, query filtering through a high-quality knowledge model, answer generation using a RAG approach, validation via refusal filtering to ensure informative responses, and knowledge boundary augmentation to address and enhance model uncertainties.

urgent need for a comprehensive pipeline focused on effectively extracting and generating large-scale QA datasets.

## 3 METHODS

Our framework involves a systematic multi-step process designed to ensure the collection of high-quality, accurate question-answer pairs, as illustrated in Fig. 1. The complete workflow is outlined as follows:

1. **Web Scraping:** We begin by batch scraping web content from sources such as Wikipedia and other informative websites. This step involves gathering extensive textual data that can be used to generate QA pairs.

2. **Query Extraction:** Subsequently, we employ prompts to extract QA pairs from the scraped web content. In this step, the answers generated by the prompts are discarded while retaining the questions. The rationale for discarding these answers is that they tend to be short and incomplete, often being merely factual entities within a paragraph. Therefore, we proceed to re-generate the answers in a more comprehensive manner in the next step.

3. **Query Filtering:** Next, we use High-Quality Knowledge Model, which is a high-quality question filter model, to extract static, clear, and knowledge-based questions.

4. **Answer Generation:** Using the retained questions as Google search queries, we retrieve the top 10+ most relevant search results. These results are then fed to a LLM through a Retrieval-Augmented Generation approach to generate detailed and accurate answers. This process results in the formation of QA pairs which are compiled as the CPT data.

5. **Validation:** This step involves a refusal filtering process, where we assess whether the generated answer contains the response to the query. This validation ensures that only those QA pairs where the answer containing valid information are included in the dataset.

6. **Knowledge Boundary Augmentation:** Finally, by utilizing the Knowledge Boundary Model to identify knowledge boundaries, we focus on areas where the model's understanding is uncertain or incomplete.

**(a) High Quality Filter Model Training Data Prompt**
*You are an expert in filtering high-quality questions. I hope you can determine whether the input question is:*
*1. clear,*
*2. static (the answer does not change within ten years), and*
*3. a knowledge-based question.*
*Here are some examples:*
*User: By whom was the Nero Decree issued on March 19, 1945?*
*Assistant: High-quality question.*
*User: On what day did the sixth season of "How to Get Away with Murder" premiere?*
*Assistant: High-quality question.*
*User: Why use seven eight for him*
*Assistant: Question is unclear/incomplete.*
*User: How is his family of origin*
*Assistant: Question is unclear/incomplete.*
*User: The volunteer activity originally limited the number of registrants to 60, increased to 80 due to high demand, explain the expansion to parents.*
*Assistant: Non-knowledge-based question.*
*User: What is the current stock price of Alibaba?*
*Assistant: Non-static question.*
*User: Linyi oil price today*
*Assistant: Non-static question.*

**(b) Refual Filtering Prompt**
*Please evaluate the provided dialogue to determine if the assistant refuses to answer the question, indicating by phrases such as:*
- *I don't know*
- *I don't want to answer*
- *Insufficient information to provide a definite answer*

*If the assistant refuses to answer, output 0; otherwise, output 1. Do not output any additional content.*

**(c) Knowledge Boundary Model Training Data Prompt**
*You are an intelligent AI tasked to determine if answers are correct or incorrect. I will provide you with a question, a standard answer, and an input answer. Based on these, please output Correct or Incorrect.*

Figure 2: Prompts used in high-quality filtering data construction: (a) High-Quality Knowledge Model training data piompt, (b) Refusal Filtering prompt, (c) Knowledge Boundary Model training data prompt.

This allows us to target and enhance specific knowledge gaps, thereby maximizing the efficiency of continuous pre-training efforts.

By following this process, we are able to curate a high-quality dataset, and it is validated as effective for enhancing the performance of CPT to improve LLM capabilities.

## 3.1 HIGH-QUALITY FILTER

In this section, we train the High-Quality Knowledge Model, which is a high-quality question filter model, to extract static, clear, and knowledge-based questions. First, we collected a batch of query-answer data. The data sources include Wiki in both Chinese and English and other online data. We annotated this data using the Qwen2.5-72B-Instruct model. The filtering prompt is engineered to achieve the following objectives:

1. **Clarity:** The input question must be clearly articulated and free from ambiguity. Ambiguous or incomplete questions can lead to incorrect or misleading responses from the model.

2. **Static Nature:** To maintain long-term knowledge representation consistency and usefulness, only questions whose answers are unlikely to change within a ten-year timeframe are considered.

3. **Knowledge-Based:** The questions must be fact-based and pertain to general knowledge rather than subjective opinions or situational contexts.

The filtering process is executed through a systematic prompt and is exemplified in Fig. 2(a). In total, we obtained 103k annotated data points, of which 46.8% were high quality, 32.6% were non-static, 13.0% were non-knowledge-based, and 7.5% were vague/incomplete. More details are in Sec. A. We divided these into training and validation sets in a 9:1 ratio and trained the High-Quality Knowledge Model based on Qwen2.5-7B model, we utilized the training dataset with the following configuration: The training was conducted on 4 GPUs with a per-device training batch size of 16. The model was trained for 1.0 epoch, with a warmup ratio of 0.03 applied at the beginning. Gradient accumulation steps were set to 4. The learning rate was initialized at $2.0 \times 10^{-6}$, and a cosine learning rate scheduler was employed.

## 3.2 REFUSAL FILTERING

In the previous sections, we have addressed the quality of queries by utilizing High-Quality Knowledge Model. Using these refined queries as inputs for Google searches, we retrieve the top 10+ most relevant search results. These results are then processed by the Qwen2.5-72B-Instruct model using the RAG approach to generate answers. Following the answer generation, we conduct a refusal filtering process, which involves reassessing all content using the Qwen2.5-72B-Instruct model to determine whether the generated answer adequately addresses the query. The assessment is guided by the prompt in Fig. 2(b). We retain only the content for which the LLM deems that the model has successfully answered the question. This validation ensures that only those QA pairs containing valid information are included in the dataset.

## 3.3 KNOWLEDGE BOUNDARY FILTERING

LLMs have demonstrated remarkable prowess in capturing and leveraging vast amounts of world knowledge through robust pre-training processes. However, simply reinforcing known knowledge during continuous pre-training offers limited benefits, primarily serving as a review and consolidation exercise. To advance these models, it is imperative to focus on areas where their understanding is uncertain or under-mastered. By identifying knowledge boundaries and enhancing specific knowledge gaps, we improve the model's performance in novel or complex scenarios, maximizing the efficiency of continuous pre-training efforts. This approach is inspired by the human learning process, where targeted practice in weaker areas leads to more comprehensive mastery of a subject.

Initially, a set of queries with known answers was compiled. For each query, we interrogated our target model using a temperature setting of 0.8, generating 30 responses. This non-greedy sampling allows for a more accurate and comprehensive evaluation of the model's grasp of specific knowledge. The generated answers were then evaluated using Qwen2.5-72B-Instruct with the prompt illustrated in Fig. 2(c) and categorized as *Correct/Incorrect*. Subsequently, we set accuracy thresholds to categorize the knowledge: queries with an accuracy greater than 0.9 were labeled as *known*, whereas those with an accuracy less than 0.1 were labeled as *unknown*. Queries with accuracy between 0.1 and 0.9 were disregarded. This strict classification standard mitigates misleading conclusions due to the uncertainty in the LLMs' outputs.

For models within the same series, while their capabilities evolve with model size and data iteration, the data they are trained on remains substantially overlapping. Thus, the known or unknown data for them likely reflects similar trends. To ensure that our knowledge boundary model is not a one-off for the pre-training process but applicable at least across different sizes and versions within the same series, we replicated this procedure with three distinct models: Qwen2-7B, Qwen2.5-7B, and Qwen2.5-72B. We aggregated the results by identifying data points with consistent labels across all three models. As shown in Fig. 3, data points marked as *unknown* by all models were tagged as *unknown*, and those marked as *known* by all models were tagged as *known*, ignoring the data points with mixed labels across the models. This consistency ensures robustness across various sizes and versions within the same model series. Consequently, this method resulted in the creation of a dataset comprising 24k training entries. The prompt used was as follows: *If you know the answer to the following question, please answer "known"; otherwise, answer "unknown".*


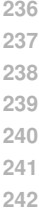
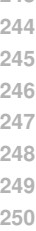

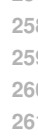
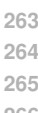
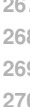

Figure 3: Knowledge boundary aggregation across model variants. Dotted lines demarcate individual models' knowledge boundaries (Qwen2-7B, Qwen2.5-7B, and Qwen2.5-72B).

|  | Chinese SimpleQA | | | | |
|---|---|---|---|---|---|
| Model | F-Score ↑ | Corr. ↑ | Incorr. ↓ | N.A. ↓ | A.A. ↑ |
| Baseline | 26.15 | 23.90 | 58.87 | 17.23 | 28.88 |
| 1×Unknown | 37.62 | 36.87 | 59.13 | **4.00** | 40.37 |
| 2×Unknown | **39.95** | **38.93** | **56.00** | 5.07 | **41.01** |

Figure 4: **Ablation results of the Knowledge Boundary Model** on Chinese SimpleQA, with metrics include Not Attempted (N.A.), Correct (Corr.), Incorrect (Incorr.), and Attempted Accuracy (A.A.). Arrows indicate the desired direction of metric improvement: ↑ for increase and ↓ for decrease. Bold indicates best results.

| Metric | Val | Chinese SimpleQA | SimpleQA |
|---|---|---|---|
| Accuracy | 95.4 | 100.0 | 99.0 |
| Precision | 94.9 | 100.0 | 100.0 |
| Recall | 94.8 | 100.0 | 99.0 |
| F1 Score | 94.8 | 100.0 | 99.5 |

Figure 5: **High-Quality Knowledge Model validation performance**. Val refers to the validation set, while ChineseSimpleQA and SimpleQA refer to the validation sets extracted from these two test datasets.

| Model | Validation A.A.(%) |
|---|---|
| Qwen2.5-72B | 81.0 |
| Qwen2.5-7B | 79.2 |
| Qwen2-7B | 82.1 |
| Unified | 83.7 |
| **Average** | 81.5 |

Figure 6: **Knowledge Boundary Model validation performance**. Unified: union where the *known/unknown* labels are consistent across all three models.

|  | Chinese SimpleQA | | | | |
|---|---|---|---|---|---|
| Model | F-Score ↑ | Corr. ↑ | Incorr. ↓ | N.A. ↓ | A.A. ↑ |
| Baseline | 26.15 | 23.90 | 58.87 | 17.23 | 28.88 |
| Fully-Finetune | 34.23 | 33.40 | 57.90 | 8.70 | 34.92 |
| HQK | **35.64** | **34.23** | **57.87** | **7.90** | **37.17** |

Figure 7: **Ablation results of the High-Quality Knowledge Model** on Chinese SimpleQA.

|  | Chinese SimpleQA | | | | |
|---|---|---|---|---|---|
| Model | F-Score ↑ | Corr. ↑ | Incorr. ↓ | N.A. ↓ | A.A. ↑ |
| Baseline | 26.15 | 23.90 | 58.87 | 17.23 | 28.88 |
| Fully-Finetune | 34.23 | 33.40 | 57.90 | 8.70 | 34.92 |
| Refusal Filtered | **36.32** | **34.97** | **57.57** | **7.87** | **37.39** |

Figure 8: **Ablation results of the refusal filtering process** on Chinese SimpleQA.

Leveraging the generated training dataset supplemented with sampled SFT data, we trained the Knowledge Boundary Model based on Qwen2.5-72B. The training process involved 3,000 iterations, with the learning rate undergoing a warm-up phase for the initial 100 iterations. A batch size of 1 was utilized, aggregated into a global batch size of 1,024 across 128 GPUs, with a gradient accumulation factor of 8. The learning rate started at $7 \times 10^{-6}$ and decayed linearly over the course of training, reaching a minimum learning rate of $7 \times 10^{-7}$. In Sec. 4.1, we validate that Knowledge Boundary Model accurately determines the knowledge boundary across models of varying parameters and versions, and enhancing the model's knowledge where it was previously unknown can effectively improve pre-training performance.

## 3.4 TRAINING PIPELINE

We propose a comprehensive framework for acquiring and curating high-quality QA datasets to enhance LLM training. Our pipeline begins with data acquisition from diverse web sources, extracting rich textual

| Model | Chinese SimpleQA | | | | | SimpleQA | | | | |
|---|---|---|---|---|---|---|---|---|---|---|
| | F-Score ↑ | Correct ↑ | Incorrect ↓ | N.A. ↓ | A.A.↑ | F-Score ↑ | Correct ↑ | Incorrect ↓ | N.A. ↓ | A.A.↑ |
| Qwen2.5-7B-Instruct | 26.15 | 23.90 | 58.87 | 17.23 | 28.88 | 3.69 | 3.14 | 67.22 | 29.63 | 4.47 |
| Our Data | 44.07 | 41.30 | 46.13 | 12.57 | 47.24 | 7.65 | 6.36 | 59.82 | 33.82 | 9.61 |
| Our Approach | **48.81** | **46.11** | **42.84** | **11.04** | **51.84** | **8.35** | **7.33** | 68.17 | 24.50 | **9.71** |

Table 1: **Performance on Simple Question-Answering** with metrics include Not Attempted (N.A.), Correct, Incorrect, and Attempted Accuracy (A.A.. Arrows indicate the desired direction of metric improvement: ↑ for increase and ↓ for decrease. Bold indicates best results.

| Model | Complex WebQuestions | Graph FreshQA | Web Questions | TruthfulQA | Question | MultiRC | TriviaQA | Avg |
|---|---|---|---|---|---|---|---|---|
| Qwen2.5-7B-Instruct | 45.87 | 43.42 | 42.01 | 62.29 | 65.27 | 30.60 | 68.80 | 51.18 |
| Our Data | 47.43 | 50.35 | 47.08 | 61.95 | 67.04 | 38.32 | 70.47 | 54.66 |
| Our Approach | **49.10** | **51.09** | **47.27** | **63.21** | **68.37** | **41.64** | **71.07** | 55.96 |

Table 2: **Performance on single-hop datasets** including ComplexWebQuestions (Talmor & Berant, 2018), FreshQA (Vu et al., 2023), GraphQuestions (Su et al., 2016), TruthfulQA (Lin et al., 2022), WebQuestions (Berant et al., 2013), MultiRC (Khashabi et al., 2018), TriviaQA (Joshi et al., 2017).

content as the foundation for QA pair generation. Next, the high-quality query extraction phase employs our High-Quality Knowledge Model to filter static, clear, and knowledge-based questions, eliminating ambiguous or unstable queries. For answer generation and validation, we combine retrieval-augmented generation with LLMs to produce detailed answers, followed by refusal filtering to ensure response validity and completeness. Finally, the knowledge boundary identification step leverages our Knowledge Boundary Model to pinpoint gaps in LLM understanding, enabling targeted training on uncertain or deficient areas. As demonstrated in Sec. 4.1, the framework's quality-control modules and training strategies significantly improve LLM capabilities.

# 4 EXPERIMENT

In this section, we first introduce the validation results of our quality control modules in Sec. 4.1 and then present the performance of our final CPT model (Sec. 4.2).

## 4.1 ABLATION

The ablation study was conducted on a subset of the training data (3.6B tokens) to validate the efficacy of our quality control modules: High-Quality Knowledge Model, refusal filtering, and Knowledge Boundary Model. The evaluation datasets used in this process include SimpleQA (Wei et al., 2024) and ChineseSimpleQA.

**High-Quality Filtering.** First, we present the validation results of our trained High-Quality Knowledge Model in Fig. 5. The model achieved an overall accuracy of 95.4% on the entire validation set. On the validation sets derived from SimpleQA and Chinese SimpleQA, due to the obviously high quality of these datasets, the classification accuracy reached over 99%. Next, we conduct ablation experiments to verify the effectiveness of the High-Quality Knowledge Model (HQK) in enhancing pre-training performance. As shown in Fig. 7, The accuracy of the model without fine-tuning is 26.2%, while the performance after complete fine-tuning is 34.2%. We perform model quality filtering on CPT data, retaining only questions that are clear,

| Model | 2Wiki MuSiQue | MultihopQA | Bamboogle | BeerQA | MultiHop CofCA | FanOutQA | FRAMES | RAG | Avg |
|---|---|---|---|---|---|---|---|---|---|
| Qwen2.5-7B-Instruct | 16.28 | 46.26 | 40.59 | 34.26 | 38.20 | 32.58 | 15.40 | 48.24 | 33.98 |
| Our Data | 22.78 | 47.50 | 42.20 | 42.83 | 47.15 | 49.12 | 33.50 | 54.99 | 42.51 |
| Our Approach | **26.21** | **49.79** | **45.83** | **43.21** | **53.76** | **50.05** | **34.06** | **56.04** | **44.87** |

Table 3: **Performance on multi-hop datasets** including MuSiQue (Trivedi et al., 2022), 2WikiMultihopQA (Ho et al., 2020), Bamboogle (Press et al., 2023), BeerQA (Qi et al., 2021), CofCA (Wu et al., 2024), FanOutQA (Zhu et al., 2024), FRAMES Krishna et al. (2025), MultiHop-RAG (Tang & Yang, 2024).

static (answers remain unchanged within ten years), and knowledge-based. The filtered data accounts for 85% of the total data. The results showed substantial gains across all measures including enhanced F-Score, increased correct rate, reduced incorrect rate, lower not attempted rate, and greater attempted accuracy. The F-Score improved by 8.1% compared to the baseline when using full data for Fully-Finetune, while a 9.5% gain was achieved with HQK model under the same training steps.

**Refusal Filtering.** We conduct ablation experiments to verify that the refusal module helps to enhance the model's pre-training performance. As shown in Fig. 8, the accuracy of the model without fine-tuning is 26.2%, and after full fine-tuning, the effectiveness improves to 34.2%. We performed refusal filtering, and the filtered data accounted for 69% of the total data. The results demonstrated significant improvements across all metrics. After applying the refusal filter, the F-score improvement rose from 8.1% for the standard dataset to 10.2%.

**Knowledge Boundary Filtering.** We present the performance of the Knowledge Boundary Model on the validation set, as shown in Fig. 6. We conducted validation using three models from the Qwen series: Qwen2.5-72B, Qwen2.5-7B, and Qwen2-7B. The Knowledge Boundary Model achieved an average classification accuracy of 80.7% across the three models. We conduct ablation experiments to validate the effectiveness of the Knowledge Boundary Model. The model labels uncertain or under-mastered knowledge as unknown. The results are presented in Fig. 4. To maintain a constant total number of training tokens, we use *1× Unknown* for the standard dataset, while *2× Unknown* increases the sampling probability of the unknown data by a factor of two. As shown in the results, the standard training achieved an F-score improvement of 11.5%, which increased to 13.8% after enhancing the unknown data.

## 4.2 PERFORMANCE

We first introduce the involved experimental configuration, followed by presenting the experimental setting and evaluation results.

### 4.2.1 EXPERIMENTAL CONFIGURATION

**Datasets.** In addition to utilizing SimpleQA (Wei et al., 2024) and ChineseSimpleQA (He et al., 2024), our study leverages a diverse array of datasets to validate the experimental results effectively. These datasets encompass both single-hop and multi-hop question-answering tasks. For single-hop tasks, ComplexWebQuestions (Talmor & Berant, 2018), FreshQA (Vu et al., 2023), GraphQuestions (Su et al., 2016), TruthfulQA (Lin et al., 2022), WebQuestions (Berant et al., 2013), MultiRC (Khashabi et al., 2018), and TriviaQA (Joshi et al., 2017) challenge models with questions requiring intricate reasoning and world knowledge. Furthermore, for more complex reasoning, multi-hop datasets such as MuSiQue (Trivedi et al., 2022), 2WikiMultihopQA (Ho et al., 2020), Bamboogle (Press et al., 2023), BeerQA (Qi et al., 2021), CofCA (Wu et al., 2024), FanOutQA (Zhu et al., 2024), FRAMES (Krishna et al., 2025), and MultiHop-RAG (Tang & Yang, 2024)

assess interconnected reasoning skills and the integration of information across multiple documents. More details are in Sec. B.

**Training Setting.** The original dataset comprises 26 billion tokens, of which 32% of low-quality data is filtered out by our filtering process. All data sources are derived from the Wiki. The dataset composition includes 20.38 billion tokens from English sources and 5.27 billion tokens from Chinese sources. The SFT phase follows the standard SFT process of Qwen. In addition to the regular SFT data, we incorporated 8,000 samples from CPT to enhance the dataset. We utilized 64 H800 GPUs for training, with the entire training process taking approximately 20 hours.

### 4.2.2 RESULTS

Our evaluation systematically examines the model's performance across three key dimensions: simple question-answering, single-hop reasoning tasks, and complex multi-hop reasoning scenarios.

**Simple Question-Answering.** Table 1 demonstrates results across Chinese SimpleQA and SimpleQA. Normal training with our data increased the F-Score from the baseline by 17.92% on Chinese SimpleQA and 3.96% on SimpleQA, validating the effectiveness of our constructed data. In contrast, our method improved the F-Score by 22.96% on Chinese SimpleQA and 4.66% on SimpleQA from the baseline. On Chinese SimpleQA, all metrics showed significant improvement, with increased accuracy, reduced error rates, and a lower proportion of unattempted questions, along with higher accuracy for attempted questions. On SimpleQA, the model showed a tendency to attempt more examples despite potential errors, leading to improvements in overall accuracy and accuracy upon attempts.

**Single-Hop Datasets.** Table 2 presents the comparative evaluation of various models on single-hop datasets. Notably, our approach consistently delivers improvements across all tested datasets, reflecting significant performance enhancements. With an overall average improvement of 3.48% and 4.78% across these datasets, our method demonstrates its efficacy in handling straightforward reasoning tasks. Specifically, on challenging tasks like MultiRC and FreshQA, our approach improved performance by 11.04% and 7.67%, respectively.

**Multi-Hop Datasets.** As illustrated in Tab. 3, our data substantially elevates performance on complex multi-hop datasets, achieving a leading average score improvement of 8.53%. Moreover, our method extends this advantage, reaching an impressive 10.89%. The model exhibits significant gains in tasks demanding intricate reasoning, such as those in the FRAMES and Bamboogle datasets, with enhancements of 18.66% and 17.47%, respectively. These results underscore the model's enhanced ability to synthesize and assess information from multiple sources, reflecting progress in executing multi-hop reasoning tasks.

## 5 CONCLUSION

In conclusion, our study mitigates the limitations in existing QA dataset creation methods by introducing a pipeline for the acquisition and filtering of QA data from diverse web sources. By leveraging the High-Quality Knowledge Model, we ensure the clarity and reliability of queries, while the Knowledge Boundary Model effectively identifies and resolves knowledge boundaries, enhancing the ability of LLMs to tackle novel challenges. The pipeline not only facilitates the generation of a substantial QA dataset but also supports advanced training strategies that significantly improve model performance.Our method achieves consistent improvements over the baseline across Chinese SimpleQA, SimpleQA, and multiple single-hop and multi-hop QA datasets, paving the way for future advancements in LLM training.

## LIMITATIONS

While our pipeline presents a novel approach to acquiring high-quality QA data, several limitations remain. First, the reliance on web-sourced content introduces variability in data quality and may inadvertently include biased or outdated information that could affect the reliability of the dataset. Although our High-Quality Knowledge Model aims to filter out such content, the dynamic nature of web sources may necessitate continuous updates and refinement of our filtering criteria. Additionally, while our framework enhances model capacity in addressing novel scenarios, its ability to reinforce unknown knowledge may vary depending on the diversity and depth of the training dataset. This underscores the importance of continuous experimentation and validation to maximize the alignment between training data and evolving model requirements. These limitations will guide future work aimed at refining our pipeline, extending its applicability, and further enhancing the performance of large language models.

## BROADER IMPACT

The development of an advanced pipeline for generating high-quality QA datasets presents significant potential to enhance Large Language Models (LLMs) as AI assistants. By improving their ability to address complex queries and identify knowledge gaps, this work contributes to advancing AI capabilities in both academic and practical settings thereby benefiting fields like education, healthcare, and customer service through more accurate and responsive interactions. However, the open-ended nature of AI technology and the vast internet data sources leveraged in this pipeline also bring potential risks of misuse. As AI models become increasingly adept at mimicking human responses, there is a risk that these systems could be used to create deceptive or manipulative content. To mitigate these risks, developers and researchers should be mindful of data biases and strive to implement ethical guidelines for responsible AI use. By doing so, we can maximize the positive impacts of this work while minimizing its potential for misuse.

## AI ASSISTANCE DISCLOSURE

This manuscript was composed with writing assistance from LLMs. After initial drafting utilizing AI tools, the authors thoroughly reviewed and refined the material, ensuring its integrity and accuracy, and assume full responsibility for the content of the final publication.

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

# A  HIGH-QUALITY KNOWLEDGE MODEL TRAINING DATASET DETAILS

The details of High-Quality Knowledge Model training dataset is shown in Tab. 4.

| Filename | Quantity | High Quality (%) | Non-static (%) | Non-knowledge -based (%) | Vague/ Incomplete (%) |
|---|---|---|---|---|---|
| Wiki CPT Data (zh) | 20000 | 50.30 | 22.56 | 11.26 | 15.88 |
| Wiki CPT Data (en) | 20000 | 88.23 | 10.15 | 0.94 | 0.69 |
| Other Online Data | 55521 | 24.07 | 48.40 | 19.92 | 7.61 |
| SimpleQA | 4226 | 100.00 | 0 | 0 | 0 |
| Chinese SimpleQA | 2900 | 100.00 | 0 | 0 | 0 |
| TOTAL | 102847 | 46.83 | 32.62 | 13.03 | 7.52 |

Table 4: High-Quality Knowledge Model training data.

# B  BENCHMARKS

We used a wide range of datasets to validate our experimental results:

1. **SimpleQA** (Wei et al., 2024) is introduced from OpenAI to assess LLMs on their ability to answer short, fact-seeking questions with a single, indisputable answer.

2. **ChineseSimpleQA** (He et al., 2024) is a comprehensive Chinese benchmark, focusing on diverse topics to test the factuality of language models in responding to concise, static questions.

3. **ComplexWebQuestions** (Talmor & Berant, 2018) is a dataset containing complex questions, including semantic parsing, search engine interaction, and reading comprehension with over 12 million web snippets.

4. **FreshQA** (Vu et al., 2023) is a dynamic QA benchmark evaluating models on questions requiring fast-changing world knowledge and debunking false premises.

5. **GraphQuestions** (Su et al., 2016) is a dataset of factoid questions paired with logical forms and ground-truth answers.

6. **TruthfulQA** (Lin et al., 2022) benchmarks language models on truthfulness, testing models with 817 questions across various domains to evaluate their mimicry of human misconceptions.

7. **WebQuestions** (Berant et al., 2013) is a popular benchmarking dataset for QA systems using structured knowledge bases, comprising 6,642 question/answer pairs.

8. **MultiRC** (Khashabi et al., 2018) is a dataset posing questions that require multi-sentence answers from diverse domains.

9. **TriviaQA** (Joshi et al., 2017) is a reading comprehension dataset with over 650K question-answer-evidence triples, demanding complex reasoning.

10. **MuSiQue** (Trivedi et al., 2022) introduces a multihop QA dataset created, featuring connected reasoning questions.

11. **2WikiMultihopQA** (Ho et al., 2020) is a multihop QA dataset using structured and unstructured data from Wikidata.

12. **Bamboogle** (Press et al., 2023) is a dataset designed to investigate language models' compositional reasoning capabilities.

13. **BeerQA** (Qi et al., 2021) is a benchmark combining existing datasets with 530 new questions requiring three Wikipedia pages to answer.

14. **CofCA** (Wu et al., 2024) introduces a Step-wise Counterfactual benchmark, revealing gaps in LLM reasoning between factual and counterfactual data.

15. **FanOutQA** (Zhu et al., 2024) presents a dataset of complex multi-hop, multi-document "fan-out" questions.

16. **FRAMES** (Factuality, Retrieval, And reasoning MEasurement Set, Krishna et al. (2025)) incorporates challenging multi-hop questions.

17. **MultiHop-RAG** (Tang & Yang, 2024) provides a knowledge base for multi-hop queries with ground-truth answers and supporting evidence, used to benchmark RAG methods.

