# OpenReview forum: "End-to-End QA Construction Pipeline for Continual Pre-training of Large Language Models"
_ICLR.cc/2026/Conference — ICLR 2026 Conference Withdrawn Submission_

### Official Review · Reviewer_XeX8 · 2025-10-30

**Soundness:** 2
**Presentation:** 2
**Contribution:** 3
**Rating:** 4
**Confidence:** 4

**Summary:**

This paper proposes a pipeline to acquire and filter high-quality QA data from web searches. The pipeline consists of (1) web scraping, (2) query extraction, (3) query filtering, (4) answer generation, (5) validation, and (6) knowledge boundary augmentation. The paper also introduces the Knowledge Boundary Model to pinpoint and address knowledge boundaries within LLMs. The proposed approach improves the baseline on Chinese SimpleQA and SimpleQA, and single/multi-hop datasets. The authors will release their 18B token filtered dataset.

**Strengths:**

- This paper presents a well-structured, multi-stage pipeline for acquiring and filtering high-quality QA data from web sources, addressing a limitation in current LLM training approaches.
- The proposed pipeline also shows consistent improvements across multiple benchmarks.
- The paper introduces a way to “interrogate” model to mine the knowledge boundary of model using a temperature of 0.8 and 30 individual responses for each query, then train a classifier to label whether a query is likely to be known or not.
- Figure 3 finding is interesting, showing that in the same model family, potentially tell us that the training data of Qwen2-7B and Qwen2.5 models are somewhat different.

**Weaknesses:**

- The entire pipeline shows heavy dependency on Qwen2.5-72B-Instruct model serving as both answer generator (step 4), self-validator (step 5), and ground truth judge (step 6). Without any external judge model or human verification, the "knowledge boundary" measures whether (smaller) models can reproduce what Qwen2.5-72B+RAG generates, rather than identifying true knowledge gaps.
- There is no validation using non-Qwen models (e.g., Llama, GPT, Claude) as either base models or evaluators, and also no human evaluation of answer correctness.
- No error analysis or failure case discussion. What type of queries get filtered?
- Knowledge Boundary Model section (3.3) is confusing and requires multiple readings, and figure 1 pipeline diagram doesn't clearly show data flow and dependencies.

**Questions:**

- Figure 2 description: typo in “piompt”
- How is accuracy calculated in section 3.3? Based on the prompt in Figure 2(c), I am assuming that we are checking the macro-average of the accuracy across the 30 responses, since the prompt asks for a binary answer. But if it is checking the token probabilities of the verbalizers then average, please make it clear.
- Is the input / training data of the knowledge boundary filtering the result of the refusal filtering?
- In section 3.3, line 228-229, are those model base model or instruct version? Are the search results given when we are using Qwen2-7B/2.5-7B/2.5-72B?
- In section 3.3, is the known answer from step 4? If that is the case, there might be a circular problem. If I understand it correctly, this is how the current pipeline like:
    1. Step 4: Answer Generation
    → Input: Question + Google search results
    → Qwen2.5-72B-Instruct generates answer A
    → This becomes the "ground truth"
    2. Step 5: Refusal Filtering
    → Qwen2.5-72B-Instruct checks: "Does my answer A address the question?"
    → If no → discard
    → If yes → keep
    3. Step 6: Knowledge Boundary Model Training
    → For each question Q with "ground truth" answer A (from step 4):
    - Given Q, ask Qwen2-7B/Qwen2.5-7B/Qwen2.5-72B for answer → get 30 responses
    - Ask Qwen2.5-72B-Instruct: "Are these responses correct?" (Comparing against A, which Qwen2.5-72B itself generated)
    - Calculate accuracy
        → If accuracy > 0.9 → known
        → If accuracy < 0.1 → unknown
        → otherwise → discard
        - Take the consistent labels across all models
   4. Step 7: Apply Knowledge Boundary Model
    → Train a Qwen2.5-72B model to predict whether new questions is "known/unknown"
    → Upsample "unknown" during experiments

    I understand that by mining the knowledge boundary mining, we save the steps of generating multiple answers per query, and instead, we can just predict based on the queries. However, the definition of ground truth seems to be circular. I hope further elaboration could be provided.

---

### Official Review · Reviewer_qodM · 2025-11-01

**Soundness:** 1
**Presentation:** 2
**Contribution:** 1
**Rating:** 2
**Confidence:** 5

**Summary:**

This paper proposes an end-to-end question-answer (QA) construction pipeline that generates QA pairs from information extracted through web search results.
The pipeline consists of a web search, QA generation, and filtering process to create high-quality QA datasets.
Experimental results show that fine-tuning models on the constructed QA pairs improves performance on SimpleQA and other related benchmarks.
The study suggests that web search–based QA generation can contribute to enhancing factual knowledge in large language models.

**Strengths:**

The paper presents a systematic pipeline for constructing question-answer pairs by integrating web search, QA generation, and filtering stages. It may be practical and scalable, enabling the creation of large QA datasets from dynamic web content. Also, the use of filtering criteria helps ensure dataset clarity and relevance.

**Weaknesses:**

**Limited novelty:**
- The idea of using web search for QA pair generation is not new—several prior works [1, 2] have already explored similar directions. However, this paper neither compares its method against those approaches nor cites them.
- Without acknowledging or improving upon earlier work, it is unclear what unique contribution this method offers.
- A discussion on the limitations of prior pipelines and how this one overcomes them would strengthen the paper.

**Ambiguous use of “high-quality”:**
- The paper claims that the generated QA pairs are high-quality, but this assertion appears overstated.
- The filtering process merely removes questions that are not clear, static, or knowledge-based, which reduces low-quality data but does not necessarily guarantee high quality.
- More rigorous criteria, such as human evaluation or factual consistency checks, would be needed to substantiate this claim.

**Weak baseline comparisons:**
- The experiments only include a limited set of baselines: pure LLM, fine-tuned LLM, and filtered-question-tuned LLM.
- Broader comparisons with existing QA-finetuned models or retrieval-augmented methods would provide stronger evidence of the proposed method’s effectiveness and novelty.

**Unclear source of performance gains:**
- Since the model’s improvement is not compared against other QA-finetuned baselines, it remains uncertain whether the observed performance gains truly stem from the proposed QA pairs.
- The enhancement could instead result from task-specific fine-tuning rather than the quality of the generated dataset itself.

[1] Wu et al., "WebWalker: Benchmarking LLMs in Web Traversal", ACL 2025.
[2] Wu et al., "WebDancer: Towards Autonomous Information Seeking Agency", NIPS 2025.

**Questions:**

1. How does the proposed QA construction pipeline differ from prior web-based QA generation methods (e.g., in data quality, diversity, or automation)?

2. Did the authors conduct any human or qualitative evaluations to assess the factual accuracy and question quality of the generated QA pairs? If not, how do they ensure that the automatically generated data truly reflects high-quality and reliable content?

---

### Official Review · Reviewer_vJhy · 2025-11-01

**Soundness:** 2
**Presentation:** 2
**Contribution:** 2
**Rating:** 2
**Confidence:** 3

**Summary:**

This paper proposes a pipeline for acquiring and filtering high-quality question-answer (QA) data from web sources to enhance continual pre-training (CPT) of large language models. The approach consists of six stages: web scraping from Wikipedia and Baidu, query extraction using LLMs, quality filtering via a trainable High-Quality Knowledge Model (HQK), answer generation through retrieval-augmented generation (RAG), validation through refusal filtering, and knowledge boundary augmentation using a Knowledge Boundary Model (KBM). The pipeline filters 26B tokens down to 18B tokens of high-quality QA data. The authors report improvements of 22.96% on Chinese SimpleQA, 4.66% on SimpleQA, 4.78% on seven single-hop datasets, and 17.47% on eight multi-hop datasets compared to baseline Qwen2.5-7B-Instruct.

**Strengths:**

The paper addresses a genuinely important problem in LLM development of creating scalable, high-quality QA datasets for continual pre-training without heavy manual annotation. The specific contribution of using learned filtering models (HQK and KBM) rather than handcrafted heuristics represents a limited but still methodological advance over prior approaches, such as Redstone. The explicit separation of clarity, staticness, and knowledge-basedness in the HQK model concerns demonstrates thoughtful problem decomposition. The knowledge boundary concept, which targets training efforts toward model uncertainties rather than reinforcing known knowledge, is conceptually sound and well-motivated by cognitive science principles of learning. The ablation studies provide evidence of the contribution of each pipeline component. The training of the KBM across three different Qwen models and aggregation to ensure consistency across model variants is a methodologically correct approach that addresses potential overfitting concerns. The evaluation is appropriate, spanning diverse QA datasets across single-hop and multi-hop reasoning tasks but Chinese and English only. The paper clearly articulates the pipeline architecture (Figure 1), provides concrete prompt examples (Figure 2), and reports detailed training configurations. The explanation of the knowledge boundary filtering by sampling responses per query and using strict thresholds demonstrates the authors' attention to reducing noise from inherent model stochasticity, but it is not analytically justified (even empirically). According to the results, the magnitude of improvements, particularly on Chinese SimpleQA (22.96%), is substantial and could be practical. However, the method is relatively straightforward, providing technical advances (that is positive) but limited understanding and knowledge of the phenomenon.

**Weaknesses:**

The KBM design raises several significant concerns. The paper uses Qwen2.5-72B-Instruct itself to label the correctness of answers. This creates a problematic dependency—the KBM is trained to recognise when a larger version of the same model family produces correct answers. This creates potential bias where model-specific reasoning patterns are reinforced rather than identifying genuinely uncertain knowledge. A more robust approach would use external ground-truth validation sets or multiple diverse evaluators. Experiments should consider across-model comparisons. The choice of thresholds lacks justification. With only 30 samples per query, these boundaries may be too rigid. A query with 27/30 correct answers (accuracy 0.9) is classified as "unknown," which seems counterintuitive. The paper would benefit from a sensitivity analysis showing how performance varies across different threshold values. There's no analysis of whether misclassified known/unknown examples actually help or harm training. The presented ablation shows improvements with 2x unknown data, but without analysing failure modes or understanding what specific types of "unknown" knowledge the model benefits from learning.

The are also data quality and filtering concerns. The paper mentions crawling from Wikipedia and Baidu but provides limited details on temporal coverage or potential staleness. For QA datasets used in evaluation (especially FreshQA), this is problematic. The "static" criterion (10-year stability) doesn't address this. It is shown that refusal filtering improves the F-score by only 1.9 percentage points despite removing 31% of the data. This modest gain raises questions about whether the filter is too aggressive or whether the removed data was genuinely of low quality. Analysis of false positives (data incorrectly filtered out) and false negatives (low-quality data retained) would strengthen this claim. The paper reports that 85% of data passes HQK filtering and 69% pass refusal filtering, but provides minimal analysis of what types of questions are filtered. Without understanding which domains, difficulty levels, or question types are removed, it's difficult to assess whether the filtering introduces systematic biases (e.g., over-representing factoid questions and under-representing opinion/ambiguous questions).

Additionally, the experiments incorporate multiple technical innovations (HQK filtering, KBM, and refusal filtering) trained on only 3.6 billion tokens for ablation. However, the full training is way bigger. The ablation doesn't explore interactions between components or validate the findings at that scale. The baseline comparison might be unfair as Qwen2.5-7B-Instruct is without CPT. A fair comparison would require training on 18B tokens of randomly selected or existing CPT data. Without this, it's unclear whether gains come from data quality specifically or simply from additional training. As I interpret the results, most gains are achieved by using more data rather than the proposed filtering.The paper provides no qualitative examples comparing baseline failures with method successes, nor does it analyze which types of questions benefit most from the approach. Understanding these patterns would strengthen claims about what the method contributes. The entire pipeline is built around and validated on Qwen models. Transferability to other model families (LLaMA, Mistral, etc.) is unvalidated. The KBM especially may encode Qwen-specific reasoning patterns. While the paper handles both Chinese and English, it's unclear how well the approach transfers to other languages, particularly low-resource ones.

This paper does not make a significant contribution to ICLR community. It makes a only practical contribution to a problem of scaling high-quality QA data acquisition for LLM training. The work suffers from methodological concerns related to the Knowledge Boundary Model's circular design, limited ablation across training scales, unfair baseline comparisons, and a lack of error analysis. The novelty of individual components is limited, though their combination is useful. The paper would be significantly strengthened by addressing the critical questions above, particularly validating that knowledge boundaries are genuinely informative rather than artifacts of the evaluation procedure, and demonstrating generalization beyond the Qwen family. With revisions addressing these concerns, this could be a strong contribution to the field

**Questions:**

1. How do you justify using Qwen2.5-72B-Instruct to evaluate answer correctness when training models in the Qwen family? Have you considered using external ground-truth sources or cross-model validation? What happens if you use a different model family (e.g., LLaMA) as the evaluator?

2. Provide ablation studies on the accuracy thresholds for known/unknown classification. How do results change with thresholds? What is the sensitivity to the threshold choice?


3. Report results for training on 18B tokens of unfiltered web-sourced QA data (not involving your filtering pipeline). This isolates the contribution of filtering from simply using more data.

4. I suggest validating that the KBM trained on Qwen models transfers to LLaMA or Mistral.

5. A much more informative experimental scenario would present the result from running full ablations (3.6B vs. 18B tokens), showing HQK only, refusal only, KBM only, and combinations. Also, validation that findings generalise from 3.6B to 18B would support the paper.

6. I would also suggest conducting a manual evaluation of 200-500 QA pairs to compare the quality of filtered vs. unfiltered pairs, and verify that "unknown" knowledge items are indeed ambiguous for the model. Report inter-rater agreement statistics.

---

### Official Review · Reviewer_jsKB · 2025-11-01

**Soundness:** 2
**Presentation:** 2
**Contribution:** 1
**Rating:** 2
**Confidence:** 4

**Summary:**

This paper proposes a new pipeline for acquiring and filtering high quality QA pairs form web searches. Authors claim that the proposed strategy not only generated good quality QA dataset but also encourages training strategies that boost LLM capabilities.  The main contributions of this paper are as follows:
1. A pipeline for QA data curation and filtering using web searches.
2. Open source release of 18B tokens QA data.
3. Experimental evaluation to show significant improvement on ChineseSimple QA , SimpleQA and other single hop and multi hop datasets.

**Strengths:**

The main strengths of the paper are as follows:
1.  The released large scale QA data might be useful for the community for research.

**Weaknesses:**

The main weaknesses of the paper are as follows:
1.  The paper has limited contributions and novelty.
2.  The QA data generation pipeline has dependence of multiple search engines for web search.
3. There are multiple web crawled corpus publicly available like common crawl, fine web etc. which could have been used to generate queries, web scraping is not required.
3. Human validation is missing, knowledge model might have its own bias and errors.
4. the paper writing has a lot of scope of improvements.

**Questions:**

1. Why the filtering step is called "high quality filtering" ? what is the prompt used for filtering?
2. Is there any comparison with existing synthetic QA dataset generation pipelines?

---

### Note · Authors · 2026-01-14

I have read and agree with the venue's withdrawal policy on behalf of myself and my co-authors.